# Osteopontin in Chronic Inflammatory Diseases: Mechanisms, Biomarker Potential, and Therapeutic Strategies

**DOI:** 10.3390/biology14040428

**Published:** 2025-04-16

**Authors:** Fuyuan Lang, Yuanheng Li, Ruizhe Yao, Meixiu Jiang

**Affiliations:** 1Queen Mary College, Jiangxi Medical College, Nanchang University, 999 Xuefu Road, Nanchang 330001, China; 4217121214@email.ncu.edu.cn (F.L.); 4217121235@email.ncu.edu.cn (Y.L.); 4217121245@email.ncu.edu.cn (R.Y.); 2The National Engineering Research Center for Bioengineering Drugs and the Technologies, Institute of Translational Medicine, Jiangxi Medical College, Nanchang University, 999 Xuefu Road, Nanchang 330031, China

**Keywords:** osteopontin, chronic inflammatory diseases, immune regulation, inflammatory signaling pathway, preclinical strategies

## Abstract

Chronic inflammatory diseases like rheumatoid arthritis, lupus, and heart disease are a major global health burden, causing long-term pain and organ damage. Scientists have discovered that a protein called osteopontin acts as a central driver of these diseases by overactivating the immune system, triggering harmful inflammation, and promoting tissue scarring. In this review, we explain how osteopontin worsens disease by recruiting immune cells, disrupting normal healing processes, and activating destructive pathways. We also highlight exciting new treatment strategies—including antibody drugs and natural plant compounds—that can block osteopontin’s harmful effects in animal studies. Importantly, measuring osteopontin levels may help doctors monitor disease severity and treatment response. Our work provides hope for millions of patients by revealing how targeting this single protein could lead to safer, more effective therapies for multiple chronic illnesses.

## 1. Introduction

Inflammation constitutes a multifaceted and protective immune response activated by deleterious stimuli, including pathogens, toxins, or physical injury [1,2]. The principal objective of this response is to eradicate the source of harm and to commence tissue repair processes [3]. Nevertheless, when inflammation persists in a chronic state, it can result in tissue dysfunction and play a contributory role in the pathogenesis of numerous diseases [4]. The inflammatory process is initiated when pattern recognition receptors (PRRs), such as Toll-like receptors (TLRs) and NOD-like receptors (NLRs), identify harmful stimuli [5]. This detection activates intracellular signaling pathways, notably NF-κB and MAPK, which subsequently induce the production of pro-inflammatory cytokines, including TNF-α and IL-1β [6]. These cytokines recruit and activate immune cells, thereby amplifying the inflammatory response. Upon activation of PRRs, signaling proteins such as the IKK complex and MAPKs initiate a cascade that facilitates the translocation of transcription factors like NF-κB and AP-1 into the nucleus [7]. This transcriptional activity leads to the synthesis of additional pro-inflammatory cytokines, such as IL-1β, TNF-α, and IL-6, which perpetuate the inflammatory response. Additionally, inflammasomes are pivotal in this process, as they activate caspase-1, resulting in the maturation and secretion of IL-1β and IL-18. Inflammasome activation can also trigger pyroptosis, a lytic form of programmed cell death mediated by the gasdermins family, like gasdermin D (GSDMD) pore formation in the cell membrane. This leads to cellular rupture and the release of additional inflammatory mediators, further amplifying immune responses and promoting immune cell recruitment to the site of injury [8]. This further propagates the inflammatory response and enhances immune cell recruitment to the site of injury.

Chronic inflammation is a characteristic feature of numerous debilitating diseases, including rheumatoid arthritis (RA), systemic lupus erythematosus (SLE), and atherosclerosis [9,10,11]. In these pathological states, elevated levels of OPN play a crucial role in exacerbating tissue damage and facilitating disease progression [12]. The global impact of inflammation-related diseases is substantial, with cardiovascular diseases alone accounting for approximately 17.9 million deaths annually [13]. This highlights the pressing need for effective management and treatment strategies for chronic inflammatory conditions. Given the central role of inflammation in the pathogenesis of a wide array of diseases, it is imperative to elucidate the molecular mechanisms involved, particularly the role of key regulatory genes, to develop targeted therapies and enhance our capacity to manage chronic inflammatory conditions effectively [14,15].

OPN, encoded by the SPP1 gene located on chromosome 4, is a multifunctional extracellular matrix protein integral to various physiological and pathological processes [16]. It is ubiquitously expressed across diverse tissues, such as bone, kidneys, immune cells, and epithelial cells, where it is pivotal in both normal physiological functions and in responses to injury or disease [17]. As a secreted protein, OPN is involved in cellular signaling and tissue remodeling, with particular significance in immune regulation, bone mineralization, and wound healing [18]. The OPN protein comprises 294 amino acids and contains multiple functional domains that underpin its diverse biological roles [19]. A prominent feature of OPN is its RGD motif (Arg-Gly-Asp), which facilitates cell adhesion by interacting with integrins, including αvβ3 and α9β1, as well as CD44 receptors [20]. This motif is critical for mediating the adhesion of immune cells and fibroblasts to extracellular matrix components, a process essential for tissue repair and immune responses [21]. Moreover, OPN exists in multiple isoforms, which are generated through alternative splicing and can be cleaved by proteases such as thrombin to yield fragments with distinct biological activities. Importantly, the interaction of OPN with integrins often requires limited proteolysis of OPN, particularly under the influence of thrombin. The thrombin-mediated cleavage of OPN at thrombin-sensitive sites like Arg168 and Ser169 in human OPN generates N- and C-terminal fragments. The N-terminal fragment contains the RGD motif, which is crucial for integrin binding, while the C-terminal fragment can interact with CD44 and other receptors [22]. Functionally, OPN is pivotal in regulating cell migration, survival, and activation [23]. It influences cellular signaling via interactions with integrins and CD44, which, upon binding, initiate several intracellular pathways, including MAPK, PI3K-Akt, and NF-Κb [24]. These signaling cascades facilitate the expression of various cytokines and chemokines, thereby modulating immune cell behavior, tissue remodeling, and fibrosis. OPN is also integral to bone remodeling processes. By binding to hydroxyapatite in bone tissue, OPN facilitates osteoclast adhesion and activation, which are crucial for bone resorption [25]. This interaction is instrumental in regulating osteogenesis and maintaining bone homeostasis. Furthermore, the capacity of OPN to influence fibrosis through the modulation of extracellular matrix deposition underscores its significance in tissue repair and pathological processes such as wound healing and organ fibrosis [26].

Given the critical role of OPN in key inflammatory pathways and its potential as a therapeutic target, a comprehensive investigation into its molecular mechanisms is imperative for understanding its full impact on chronic inflammation and associated diseases. This review examines the significant role of OPN in modulating inflammation across various immune cells, including macrophages, neutrophils, and T cells. OPN exerts its influence by modulating key inflammatory signaling pathways such as NF-κB, MAPK, and PI3K/Akt, thereby affecting immune cell activation, recruitment, and cytokine secretion. We also provide an overview of the involvement of OPN in a spectrum of chronic inflammatory diseases, including rheumatoid arthritis, atherosclerosis, psoriasis, and inflammatory bowel disease, where its expression is correlated with disease severity and progression. Additionally, we evaluate the therapeutic potential of targeting OPN in these diseases, offering insights into how OPN inhibition could present novel strategies for managing chronic inflammation. The subsequent sections will explore the molecular mechanisms by which OPN regulates inflammation, its role in specific diseases, and the clinical implications of targeting OPN as a therapeutic strategy.

To ensure a comprehensive and balanced narrative review, we conducted a literature search using multiple databases, including PubMed, Scopus, and Web of Science. We focused on peer-reviewed articles published in the past decade, prioritizing studies that investigated the molecular mechanisms of OPN, its role in immune regulation, and its relevance to chronic inflammatory diseases. Studies were included based on their relevance to OPN signaling, inflammation, and disease models, while articles lacking mechanistic insights or those with limited experimental validation were excluded. Despite these efforts, certain limitations must be acknowledged, such as potential publication bias, variations in experimental models across studies, and the need for further validation in clinical settings. Nevertheless, this review aims to provide a comprehensive synthesis of current knowledge on OPN, highlighting its role as a key regulator of inflammation and a potential therapeutic target.

## 2. The Biological Functions of OPN

### 2.1. OPN Regulates Immune Cell Recruitment and Activation

OPN is integral to the recruitment and activation of immune cells, particularly within the context of inflammatory processes [27] (Table 1). It facilitates the mobilization of monocytes, macrophages, T cells, and dendritic cells (DCs) through its interactions with various cell surface receptors, including integrins (αvβ3 and αvβ5) and CD44. These receptor-mediated interactions promote the adhesion of immune cells to endothelial cells, facilitating their transmigration across the vasculature and subsequent infiltration into inflamed tissues [28] (Figure 1).

#### 2.1.1. OPN and Monocyte Recruitment

Monocytes play a pivotal role in the initial immune response [29]. OPN facilitates the recruitment of monocytes to sites of inflammation by binding to integrins (αvβ3 and α4β1) present on the monocyte surface [30]. This interaction enhances the adhesion of monocytes to endothelial cells, a crucial step in their transmigration into tissues. Additionally, OPN engages with CD44, thereby promoting monocyte motility and facilitating tissue infiltration [39]. Upon recruitment to inflammatory sites, monocytes differentiate into macrophages, which are integral to the immune response through the phagocytosis of pathogens and clearance of tissue debris [40].

Moreover, OPN augments the production of chemokines such as CCL2, which further recruit monocytes to the inflammatory locus. This chemotactic activity is vital for an effective immune response, as it amplifies monocyte infiltration, thereby playing a crucial role in resolving infection or tissue injury.

#### 2.1.2. OPN and Macrophage Activation

Macrophages are integral to both the initiation and resolution of inflammation [41]. Upon differentiation from monocytes, macrophages are activated and polarized towards the M1 phenotype by OPN, which is marked by the secretion of pro-inflammatory cytokines such as tumor necrosis factor-alpha (TNF-α), interleukin-1 beta (IL-1β), and interleukin-6 (IL-6) [31]. These cytokines facilitate the recruitment of additional immune cells and perpetuate the inflammatory response. OPN exerts its effects through interactions with integrins and CD44, enhancing macrophage adhesion, migration, and functional capabilities in pathogen recognition and cytokine production [42,43,44]. Furthermore, OPN supports the M1 polarization of macrophages, ensuring a potent inflammatory response necessary for pathogen clearance, although it may lead to tissue damage if not properly regulated [45]. OPN also influences macrophage metabolism, particularly by regulating glycolysis, which is essential to meet the energy demands of activated macrophages [32]. This metabolic reprogramming enables macrophages to generate reactive oxygen species (ROS) and inflammatory cytokines, which are critical for their inflammatory functions [33,46].

#### 2.1.3. OPN and T Cell Activation

OPN plays a pivotal role in the activation and differentiation of T cells, particularly Th1 and Th17 subsets, which are integral to the mediation of inflammatory and autoimmune responses [34,47]. In Th1 cells, OPN enhances the expression of interferon-gamma (IFN-γ), a cytokine that activates macrophages and further intensifies the inflammatory response [48]. IFN-γ also contributes to the augmentation of macrophage function, thereby reinforcing the immune response at sites of infection or injury [35]. Moreover, OPN is implicated in the differentiation of Th17 cells, which produce interleukin-17 (IL-17), a cytokine that recruits neutrophils and exacerbates inflammation [49]. Through its regulation of Th1 and Th17 differentiation, OPN ensures the sustained activation of adaptive immunity during chronic inflammatory conditions [50,51].

Additionally, OPN facilitates the migration of T cells into inflamed tissues by promoting their adhesion to the endothelium via interactions with integrins and CD44 [52]. This capacity to enhance T cell trafficking ensures efficient recruitment to sites of inflammation, where T cells can perform their immune functions, including cytokine production and pathogen clearance.

#### 2.1.4. OPN and Dendritic Cell Recruitment and Activation

Dendritic cells (DCs) play a pivotal role in initiating adaptive immune responses by capturing and presenting antigens to T cells [36]. OPN facilitates the recruitment and activation of DCs through interactions with integrins and CD44, thereby promoting their migration into inflamed tissues [37,53]. Upon reaching the site of infection or injury, OPN enhances the antigen-presenting capabilities of DCs and their ability to activate T cells [54]. This process is essential for the initiation of adaptive immunity, as DCs are responsible for priming naïve T cells and tailoring the immune response to specific pathogens or injuries [55]. Furthermore, OPN contributes to the retention of DCs within inflamed tissues by promoting their adhesion to the extracellular matrix [56]. This retention enables DCs to efficiently process and present antigens to T cells, thereby fostering a robust and localized immune response.

#### 2.1.5. OPN and Immune Cell Crosstalk

The recruitment and activation of immune cells by OPN are not restricted to specific cell types [57]. OPN-activated macrophages and T cells engage in intercellular communication via cytokines and chemokines, thereby augmenting the immune response [38]. For instance, macrophages stimulated by OPN secrete interleukin-6 (IL-6), tumor necrosis factor-alpha (TNF-α), and chemokine (C-C motif) ligand 2 (CCL2), which facilitate the activation and recruitment of T cells [58]. This interaction between macrophages and T cells intensifies the inflammatory response, ensuring the immune system’s capacity to effectively target and eliminate pathogens [59]. Furthermore, OPN has been implicated in the regulation of regulatory T cell (Treg) function, which is crucial for modulating immune responses and maintaining immune tolerance [60]. In the context of chronic inflammation, however, OPN may impair Treg function, resulting in persistent inflammation.

### 2.2. OPN Modulates Cytokine Expression to Drive Inflammation

OPN is pivotal in the regulation of inflammation through its modulation of both pro-inflammatory and anti-inflammatory cytokine expression [61]. OPN augments the production of critical pro-inflammatory cytokines, such as tumor necrosis factor-alpha (TNF-α), interleukin-6 (IL-6), and interleukin-1 beta (IL-1β), thereby contributing to the amplification of the immune response [62]. TNF-α, one of the initial cytokines produced in response to injury or infection, is upregulated by OPN in macrophages and other immune cells, thereby intensifying the inflammatory response and facilitating the recruitment of additional immune cells to the inflammation site [63]. Similarly, OPN increases IL-6 production, a cytokine integral to the acute-phase response, which assists in the recruitment and differentiation of T helper (Th) 17 cells [64]. IL-6 also stimulates IL-17 secretion from Th17 cells, further exacerbating inflammation in autoimmune diseases. Furthermore, OPN enhances the production of IL-1β, which is crucial in initiating the inflammatory cascade and promoting neutrophil recruitment to the site of injury.

OPN not only promotes pro-inflammatory cytokines but also suppresses the expression of the anti-inflammatory cytokine interleukin-10 (IL-10), which is crucial for resolving inflammation and restoring immune homeostasis [65]. OPN inhibits IL-10 production in macrophages and other immune cells, thereby impairing the resolution of inflammation and facilitating the persistence of chronic inflammatory states [66]. By inhibiting IL-10, OPN perpetuates unchecked inflammation, leading to the sustained activation of T helper 1 (Th1) and T helper 17 (Th17) cells, both of which are integral to the pathogenesis of chronic inflammation and autoimmune diseases such as rheumatoid arthritis and atherosclerosis [67]. This suppression of IL-10 contributes to prolonged immune system activation and exacerbates tissue damage.

The pro-inflammatory effects of OPN are further potentiated through its activation of intracellular signaling pathways, notably NF-κB and mitogen-activated protein kinases (MAPKs) such as p38, extracellular signal-regulated kinase (ERK), and c-Jun N-terminal kinase (JNK) [68]. NF-κB, a pivotal transcription factor, is activated by OPN via receptor-mediated interactions, resulting in its translocation to the nucleus and subsequent induction of pro-inflammatory cytokine production, including tumor necrosis factor-alpha (TNF-α) and interleukin-6 (IL-6) [69]. The MAPK signaling cascade also plays a crucial role in OPN-mediated cytokine expression, particularly through the p38 and ERK pathways, which facilitate the transcription of inflammatory mediators by activating activator protein-1 (AP-1). These signaling pathways collectively augment the production of inflammatory cytokines and chemokines, thereby amplifying the immune response and promoting the recruitment of immune cells to the site of inflammation.

To sum, OPN modulates the equilibrium between pro-inflammatory and anti-inflammatory cytokines by enhancing the production of inflammatory cytokines such as TNF-α, IL-6, and IL-1β while concurrently suppressing IL-10, thereby establishing a chronic pro-inflammatory milieu [70]. This imbalance in cytokine production underscores involvement in the pathogenesis of chronic inflammatory diseases and autoimmune disorders, where its overactivation perpetuates inflammation and tissue damage [71].

### 2.3. OPN Regulates Cell Survival and Death

OPN serves as a critical regulator of cellular survival and death (Figure 2), modulating the equilibrium between cell viability and death [72] (Table 2). Its function in apoptosis is context-dependent, and while OPN is widely recognized as an anti-apoptotic factor, its effects can be ambiguous and vary across different cell types and conditions. By interacting with its surface receptors, OPN activates intracellular signaling pathways, notably the PI3K/Akt pathway, which are essential for enhancing cell survival and suppressing apoptotic processes [73]. Through the engagement of these signaling cascades, OPN enables cells to withstand apoptotic stimuli and extends their viability, particularly under stress conditions such as inflammation or tissue injury [74]. This survival-promoting mechanism is crucial for maintaining tissue homeostasis and supporting the immune response. Empirical studies have demonstrated that OPN can activate the PI3K/Akt pathway, resulting in the phosphorylation of Akt [75]. This phosphorylation event subsequently inhibits pro-apoptotic proteins, including Bad and Bax, while augmenting the activity of anti-apoptotic proteins such as Bcl-2 [76]. In cells experiencing stress or inflammation, the activation of the PI3K/Akt pathway by OPN is instrumental in preserving mitochondrial membrane integrity, which is vital for preventing the initiation of the mitochondrial apoptotic pathway. The stabilization of mitochondria by OPN ensures the inhibition of pro-apoptotic signals, thereby extending cellular longevity. By inhibiting caspase activation and reducing the mitochondrial release of cytochrome c, OPN effectively prevents apoptosis in cells under inflammatory stress [77]. This protective role of OPN is important not only for immune cells but also for other cell types in inflamed or injured tissues, where it helps preserve functional cells and aids in tissue repair processes.

Moreover, the capacity of OPN to regulate apoptosis extends beyond the PI3K/Akt signaling pathway [78]. OPN also activates additional survival-related signaling pathways, notably the mitogen-activated protein kinase (MAPK) pathway, which encompasses extracellular signal-regulated kinase (ERK), c-Jun N-terminal kinase (JNK), and p38. These pathways contribute to cellular survival by promoting the expression of genes that inhibit apoptosis and support cell proliferation. For instance, OPN enhances ERK activation, which subsequently elevates the expression of proteins integral to cell cycle progression and survival, such as Cyclin D1 and Bcl-2 [79]. This interplay between the PI3K/Akt and MAPK pathways amplifies the survival-promoting effects of OPN, thereby ensuring prolonged cell survival in damaged tissues and facilitating repair and regeneration.

However, the role of OPN in apoptosis is not unidirectional. In certain circumstances, OPN can also promote cell death, especially under specific stress conditions or in the context of pathological inflammation. Moreover, OPN is implicated in pyroptosis, a form of inflammatory programmed cell death mediated by inflammasome activation. By modulating NLRP3 inflammasome activity, OPN can regulate caspase-1 activation and gasdermin family-mediated membrane pore formation, thereby influencing inflammatory cell death in chronic inflammatory diseases [80]. In addition to regulating intrinsic apoptosis pathways, OPN also regulates exogenous apoptosis mediated by death receptor signaling. OPN has been reported to affect Fas/FasL and TNF receptor-related apoptosis, making cells sensitive to death receptor-induced apoptosis. For instance, evidence indicates that in cardiovascular tissues, OPN can enhance apoptosis in cardiomyocytes under ischemic stress, possibly through interactions with death receptor pathways and mitochondrial dysfunction [81]. Additionally, OPN interacts with ER stress pathways, influencing CHOP and ATF4 signaling, which can determine whether cells undergo apoptosis or adaptive responses under stress conditions [82]. Thus, while OPN generally functions as an anti-apoptotic factor, its effects are highly context-dependent, with the potential to either protect or promote cell death depending on the specific inflammatory and stress-related environment.

In addition to its impact on immune cells, OPNs have been demonstrated to affect the survival of non-immune cells, including fibroblasts and epithelial cells, particularly under stress conditions [83]. In fibroblasts, OPN facilitates cell survival via the PI3K/Akt signaling pathway, which upregulates the expression of survival factors and inhibits apoptosis triggered by inflammatory mediators. This anti-apoptotic effect is pivotal in fibrotic diseases, where extended cell survival can contribute to pathological tissue remodeling. Similarly, in epithelial cells, OPN activates survival pathways that enable these cells to endure inflammatory stress, which is essential for preserving the integrity of epithelial barriers [84].

**Table 2 biology-14-00428-t002:** Mechanisms regulating death and survival pathways in OPN-mediated chronic inflammation.

Survival/Death	Influencing Pathways/Factors	Mechanism	References
Cell survival	PI3K/Akt pathway	Activates Akt, inhibits pro-apoptotic proteins (Bad, Bax), and promotes anti-apoptotic proteins (Bcl-2, Bcl-xL), enhancing cell survival	[74,75]
	Mitochondrial integrity	Stabilizes mitochondrial membranes, prevents cytochrome c release, and inhibits caspase-9 activation to suppress apoptosis	[76]
	MAPK pathway (ERK, JNK, p38)	Activates ERK, JNK, and p38, promoting survival by upregulating Cyclin D1 and Bcl-2, while JNK may have context-dependent pro-apoptotic effects	[78]
	Survival factors (fibroblasts, epithelial cells)	Enhances survival by upregulating protective factors such as VEGF, HIF-1α, and STAT3 to maintain cell viability under stress conditions	[79]
Cell death	Pyroptosis (gasdermin D-mediated)	OPN influences inflammasome activation (e.g., NLRP3), promoting or inhibiting pyroptosis via caspase-1 and gasdermin D cleavage	[80]
	ER stress and autophagy interactions	OPN modulates ER stress responses and autophagy, which can either promote survival or trigger apoptosis through CHOP and ATF4 pathways	[81]
	Death receptor pathways (Fas/FasL, TNF-α, TRAIL)	Modulates extrinsic apoptosis by regulating Fas and TNFR signaling, either inhibiting or sensitizing cells to apoptosis depending on cellular context	[82]
	Caspases, mitochondrial signals	Inhibits caspase activation and mitochondrial apoptotic signals, but in some contexts, excessive OPN may promote apoptosis via p53-dependent pathways	[84]

Abbreviations: OPN: osteopontin; PI3K/Akt: phosphatidylinositol 3-kinase/protein kinase B; MAPK: mitogen-activated protein kinase; ERK: extracellular signal-regulated kinase; JNK: c-Jun N-terminal kinase; NLRP3: NOD-like receptor family pyrin domain-containing 3; GSDMD: Gasdermin D; ER: Endoplasmic Reticulum; ATF4: Activating transcription factor 4; Fas: Fas cell surface death receptor; FasL: Fas ligand; TNFR: Tumor necrosis factor receptor; TNF-α: Tumor necrosis factor alpha; TRAIL: TNF-related apoptosis-inducing ligand; p53: Tumor protein p53; Caspase-1: Cysteine-aspartic protease 1. References in the table correspond to studies cited in the main text.

### 2.4. OPN Involves in Tissue Remodeling and Fibrosis

OPN serves as a critical mediator in tissue remodeling and fibrosis, especially within the context of chronic inflammation [85]. It orchestrates various cellular processes that facilitate the remodeling of extracellular matrix (ECM) components, notably through the proliferation of fibroblasts and the synthesis of collagen. In tissues undergoing repair or inflammation, OPN functions as a potent stimulator of fibroblasts, which are the primary cells responsible for ECM production, particularly collagen [16]. OPN promotes fibroblast activation by binding to its surface receptors, thereby initiating several intracellular signaling pathways, including PI3K/Akt and MAPK. These pathways contribute to the enhanced proliferation of fibroblasts, thereby enabling their participation in tissue repair and fibrosis. A pivotal function of OPN in the pathogenesis of fibrosis is its regulation of collagen synthesis. Empirical evidence indicates that OPN augments the production of collagen type I, the predominant constituent of the fibrotic matrix, via fibroblasts [86]. This upregulation of collagen synthesis is linked to the activation of Transforming Growth Factor-beta (TGF-β), a pivotal cytokine implicated in fibrotic processes. OPN facilitates TGF-β signaling, thereby enhancing collagen gene expression in fibroblasts, which contributes to the accumulation of extracellular matrix (ECM) components and the advancement of fibrosis [87]. Furthermore, OPN elevates the expression of additional fibrogenic markers, such as fibronectin, which plays a role in ECM assembly and the stabilization of the fibrotic matrix. In addition, in chronic inflammatory conditions, including liver fibrosis, pulmonary fibrosis, and chronic kidney disease, OPN expression is significantly upregulated as a response to sustained injury and inflammation. This overexpression of OPN exacerbates the fibrotic process by perpetuating fibroblast activation and promoting excessive ECM deposition [88]. Specifically, in liver fibrosis, OPN facilitates the activation of hepatic stellate cells, a critical step in the fibrotic response [89]. OPN interacts with these cells to enhance their proliferation and collagen production, resulting in the thickening of fibrotic scar tissue and subsequent liver dysfunction. Similarly, in pulmonary fibrosis, OPN modulates the equilibrium between ECM degradation and synthesis, thereby favoring collagen accumulation and the progression of fibrosis [90].

OPN plays a pivotal role in tissue remodeling, not only by promoting ECM production but also by regulating ECM turnover [91]. OPN modulates the activity of matrix metalloproteinases (MMPs), key enzymes involved in ECM degradation. The dysregulated expression of OPN can lead to suppressed MMP activity, thereby hindering ECM breakdown and facilitating the accumulation of fibrotic tissue. This disruption results in an imbalance between ECM synthesis and degradation, driving fibrosis and compromising tissue function [92]. Chronic inflammation, often associated with excessive OPN production, creates a microenvironment in which fibroblasts are persistently activated, leading to continuous ECM synthesis. This overactive fibrotic response contributes to fibrosis in various organs, impairing their normal architecture and function [93]. These findings highlight the essential role of OPN in pathological tissue remodeling, particularly in the context of chronic inflammatory diseases.

To summarize, OPN plays a crucial role in the regulation of immune cell recruitment and activation, cytokine production, and tissue remodeling, making it a key mediator in inflammatory responses. By interacting with various receptors, OPN promotes the migration and activation of monocytes, macrophages, T cells, and dendritic cells, thereby enhancing local inflammation. Furthermore, its involvement in regulating cell survival, apoptosis, and fibrosis exacerbates tissue damage and chronic inflammation. Given its widespread involvement in these processes, the role of OPN extends beyond immune modulation to influence the tissue microenvironment, accelerating the progression of chronic inflammatory diseases such as rheumatoid arthritis, systemic lupus erythematosus, and chronic kidney disease.

## 3. The Multifaceted Role of OPN in Inflammatory Disease Progression

OPN is integral to the progression of various inflammatory diseases, significantly affecting immune cell recruitment, cytokine production, and tissue remodeling. In pathologies such as rheumatoid arthritis, systemic lupus erythematosus, and chronic kidney disease, elevated levels of OPN facilitate immune cell activation and fibroblast proliferation, thereby promoting extracellular matrix deposition and fibrosis (Figure 3). This cascade exacerbates inflammation and tissue damage, contributing to the chronic progression and increased severity of these diseases (Table 3). Moreover, the activation of signaling pathways, including TGF-β and NF-κB, perpetuates a persistent inflammatory microenvironment that sustains disease chronicity. Through its multifaceted roles, OPN integrates immune responses with structural tissue remodeling, establishing itself as a central mediator in the pathology of inflammatory diseases. The subsequent sections will elaborate on the impact of OPN across specific inflammatory conditions, offering insights into its distinct molecular mechanisms and influence on disease progression. However, although OPN has been widely recognized as a pro-inflammatory mediator due to its role in immune cell recruitment, cytokine induction, and inflammasome activation, its function in inflammation regulation is not entirely unidirectional. Recent evidence suggests that OPN deficiency can paradoxically contribute to low-grade chronic inflammation, possibly due to impaired immune regulation and defective tissue repair mechanisms [94]. For example, while OPN promotes inflammatory responses in autoimmune diseases such as rheumatoid arthritis (RA) and systemic lupus erythematosus (SLE), its absence may lead to the inadequate resolution of inflammation, persistent tissue damage, and low-grade chronic inflammation. This effect may be mediated by dysregulated macrophage function, impaired clearance of apoptotic cells, and altered cytokine homeostasis. Furthermore, OPN-deficient models have demonstrated increased susceptibility to metabolic and fibrotic disorders associated with chronic low-grade inflammation, underscoring its role in immune homeostasis rather than merely serving as a pro-inflammatory driver.

### 3.1. OPN and Rheumatoid Arthritis

Rheumatoid arthritis (RA) is a chronic autoimmune condition predominantly impacting the joints, characterized by ongoing synovial inflammation, joint degradation, and potential systemic complications [111]. The onset and progression of RA result from a multifaceted interaction among genetic predisposition, environmental factors, and immunological disruptions, culminating in an aberrant immune response and sustained inflammation in the affected tissues. This pathological immune activity is perpetuated by pro-inflammatory cytokines, autoantibodies, and immune cells infiltrating the synovium, which leads to synovial hyperplasia and the formation of destructive pannus tissue, causing irreversible damage to cartilage and bone. Significantly, increased levels of OPN have been detected in the serum and synovial fluid of RA patients, with a strong correlation to disease activity and severity. OPN plays an active role in RA pathogenesis by promoting synovial fibroblast proliferation and enhancing the production of pro-inflammatory cytokines such as TNF-α, IL-1β, and IL-6, thereby exacerbating synovial inflammation and expediting joint destruction [111]. Moreover, OPN plays a pivotal role in the recruitment and activation of immune cells, such as macrophages and T cells, within synovial tissue, thereby intensifying the inflammatory environment and sustaining the chronic inflammation characteristic of rheumatoid arthritis (RA) [112]. Experimental studies using murine models highlight the significant role of OPN in RA, as OPN-deficient mice demonstrate a marked reduction in the severity of collagen-induced arthritis, indicating the importance of OPN in driving disease progression [113]. Preclinical efforts to target OPN have shown promise, with evidence suggesting that OPN inhibition attenuates inflammatory responses and reduces joint destruction, reinforcing its potential as a biomarker and therapeutic target in RA [16]. By modulating immune cell activity, cytokine production, and synovial cell proliferation, OPN emerges as a central factor in perpetuating the inflammatory and degenerative pathology of RA.

### 3.2. OPN and Systemic Lupus Erythematosus

Systemic lupus erythematosus (SLE) is a chronic autoimmune disorder characterized by widespread inflammation affecting multiple organs, including the kidneys, skin, joints, and central nervous system [95]. This condition is primarily driven by immune dysregulation, particularly the aberrant activation of B and T cells, leading to the production of autoantibodies that form immune complexes. These complexes deposit in various tissues, triggering inflammation and tissue damage. In SLE patients, elevated serum OPN levels are consistently observed and correlate strongly with disease activity and the degree of organ damage [114]. OPN plays a critical role in the pathogenesis of SLE by enhancing B cell activation, which in turn increases the production of autoantibodies, further fueling the autoimmune response [96]. Additionally, OPN promotes the differentiation of plasmacytoid dendritic cells (pDCs), which are instrumental in the production of type I interferons (IFN-I), a key cytokine family involved in the inflammatory processes of SLE [97]. This OPN-induced IFN-I response establishes a positive feedback loop that sustains chronic inflammation and drives the expansion of autoreactive T cell populations. These T cells amplify the release of pro-inflammatory cytokines, exacerbating tissue damage and accelerating the progression of organ-specific complications, such as lupus nephritis and neuropsychiatric involvement [115]. Both clinical and experimental studies have shown that elevated OPN levels can serve as reliable predictors of disease progression and severity, suggesting its potential as a biomarker for monitoring SLE activity [116]. Given its central role in immune regulation and disease progression, targeting OPN offers significant therapeutic promise for modulating immune responses and alleviating disease severity in SLE.

### 3.3. OPN and Multiple Sclerosis

Multiple sclerosis (MS) is a chronic autoimmune disorder affecting the central nervous system (CNS), characterized by inflammation, demyelination, and axonal damage, which collectively lead to progressive neurodegeneration and a broad spectrum of symptoms, including motor dysfunction, cognitive impairment, and sensory disturbances [117]. Elevated levels of OPN are detectable in both the cerebrospinal fluid (CSF) and serum of individuals with MS, with a strong correlation observed between OPN concentration and disease severity [118]. OPN is integral to MS pathogenesis, as it promotes T cell activation and facilitates the differentiation of Th1 and Th17 cells, which are critical mediators of CNS inflammation and myelin degradation. Research utilizing experimental autoimmune encephalomyelitis (EAE), a well-established animal model for MS, indicates that OPN deficiency leads to a significant reduction in disease severity, thereby underscoring the role of OPN in driving MS progression [119]. In OPN-deficient mice, the absence of this protein substantially mitigates symptom severity, highlighting its contribution to the inflammatory processes underlying MS [27]. Furthermore, elevated OPN levels are associated with increased production of pro-inflammatory cytokines, including IFN-γ and IL-17, which perpetuate immune responses and exacerbate CNS tissue damage [48]. Through its modulation of T cell activity and enhancement of cytokine production, OPN not only amplifies inflammatory responses but also supports the survival and persistence of autoreactive T cells, thereby exacerbating MS pathology and accelerating neurodegeneration. Given these insights, OPN is considered a potential biomarker for monitoring MS disease activity and a promising therapeutic target aimed at reducing CNS inflammation and halting disease progression [98].

### 3.4. OPN and Inflammatory Bowel Disease

Inflammatory bowel disease (IBD), which includes Crohn’s disease and ulcerative colitis, is a chronic, relapsing condition of the gastrointestinal tract resulting from a dysregulated immune response to intestinal microbiota [120]. IBD is marked by persistent inflammation that disrupts the intestinal mucosal barrier, leading to symptoms such as abdominal pain, diarrhea, weight loss, and nutrient deficiencies, with patients often experiencing cycles of remission and relapse [121]. Elevated OPN levels have been detected in the intestinal mucosa of IBD patients, and its expression strongly correlates with disease activity. OPN plays a critical role in maintaining the intestinal barrier by promoting epithelial cell survival and enhancing mucosal regeneration, which is vital for counteracting tissue damage from chronic inflammation and preserving gut integrity [122].

However, excessive OPN expression can exacerbate inflammation by recruiting immune cells, particularly macrophages and neutrophils, to the intestinal lining, thus intensifying the inflammatory response [99]. This immune cell recruitment facilitates the release of pro-inflammatory cytokines, including TNF-α and IL-6, which perpetuate inflammation and hinder normal tissue repair processes. Chronic upregulation of these cytokines not only impairs epithelial recovery but also contributes to fibrosis, a frequent complication of prolonged IBD [100]. OPN has been shown to promote fibroblast activation and collagen synthesis, leading to excessive scarring and the formation of strictures that impair intestinal motility and function [86]. Experimental models of colitis further support the involvement of OPN in IBD pathogenesis, as OPN deficiency results in reduced inflammation, suggesting that targeting OPN could offer a potential strategy to modulate the inflammatory cascade and alleviate disease severity of IBD.

Given its dual role in supporting epithelial repair and perpetuating inflammation, OPN has emerged as both a valuable biomarker for monitoring IBD activity and a promising therapeutic target. Modulating OPN expression offers a potential therapeutic strategy to reduce inflammation, mitigate fibrosis, and ultimately improve long-term outcomes in patients with IBD.

### 3.5. OPN and Atherosclerosis

Atherosclerosis is a chronic, progressive vascular disease characterized by the accumulation of lipids, inflammatory cells, and smooth muscle cells within the arterial walls, resulting in plaque formation that narrows the arteries and restricts blood flow [101]. This pathological process increases the risk of serious cardiovascular events, such as myocardial infarction, stroke, and peripheral artery disease. The disease is initiated by endothelial injury, which triggers an inflammatory response and attracts immune cells to the site of damage. Over time, this leads to the formation of a fibrous plaque containing a lipid-rich core, a hallmark feature of atherosclerosis [123].

OPN is a key mediator in the pathogenesis of atherosclerosis, with high expression levels observed in atherosclerotic plaques, particularly in macrophages and vascular smooth muscle cells (VSMCs) [124]. OPN has been shown to play a critical role in both plaque formation and its instability. Elevated OPN levels correlate with increased plaque vulnerability, serving as a strong predictor of adverse cardiovascular outcomes. Its contribution to atherosclerosis is multifaceted, involving both inflammatory and fibrotic processes that promote plaque development and destabilization.

One of the primary functions of OPN in atherosclerosis is the recruitment and activation of macrophages. Upon migration to the arterial walls, macrophages differentiate into foam cells, a critical event in plaque formation [125]. These foam cells accumulate lipids, forming the lipid-rich core of the plaque, a feature strongly associated with unstable plaques that are prone to rupture. Furthermore, OPN enhances the uptake of modified low-density lipoproteins (LDLs) and other lipoproteins by macrophages, thereby accelerating foam cell formation and promoting the expansion of the plaque [102]. As a result, plaques grow larger and more complex, increasing their risk of rupture, which can lead to life-threatening cardiovascular events.

In addition to its role in macrophage activation and foam cell formation, OPN also influences the behavior of VSMCs, which are responsible for the formation of the fibrous cap that stabilizes atherosclerotic plaques. While VSMC migration and proliferation are essential for the initial stabilization of plaques, excessive OPN expression can lead to abnormal VSMC proliferation and migration, contributing to pathological vascular remodeling [103]. This process thickens the arterial wall and further narrows the vessel lumen, exacerbating atherosclerosis and increasing arterial stiffness. These changes significantly impair blood flow and further promote the progression of the disease.

The dual roles of OPN in both promoting plaque formation and contributing to plaque instability highlight its central role in atherosclerosis. Through its actions on macrophage recruitment, foam cell formation, and VSMC remodeling, OPN is deeply involved in the inflammatory, fibrotic, and structural processes that drive plaque progression and destabilization. Elevated OPN levels are not only associated with increased plaque vulnerability but are also considered a potential biomarker for assessing the risk of acute cardiovascular events [104]. Animal models have demonstrated that OPN deficiency reduces plaque size and inflammatory cell infiltration, providing compelling evidence that targeting OPN may help reduce plaque burden and improve cardiovascular outcomes [126].

Given its critical involvement in both plaque formation and destabilization, OPN represents an attractive therapeutic target for modulating atherosclerotic progression. Strategies aimed at inhibiting OPN or its receptor interactions could help limit macrophage infiltration, foam cell formation, and VSMC migration, potentially stabilizing plaques and preventing rupture. Additionally, as OPN plays a role in the fibrotic remodeling of arterial walls, therapies targeting OPN may mitigate vascular remodeling and reduce the risk of complications associated with plaque rupture. Beyond its therapeutic potential, the presence of OPN in circulation as a biomarker offers a valuable tool for assessing the severity of atherosclerosis, enabling better risk stratification and more informed clinical decision-making. In summary, OPN not only provides crucial insights into the mechanisms underlying atherosclerosis but also holds promise as a target for therapeutic interventions aimed at reducing cardiovascular morbidity and mortality.

### 3.6. OPN and Chronic Kidney Disease

Chronic kidney disease (CKD) is a progressive condition characterized by a gradual decline in kidney function, which can ultimately result in end-stage renal disease (ESRD). Often accompanied by comorbidities such as hypertension and diabetes, CKD involves persistent damage to kidney structures, particularly the glomeruli and tubules. As the disease progresses, kidney function deteriorates due to structural changes and increased fibrosis [127]. In CKD patients, OPN levels are significantly elevated in both serum and urine, and its expression correlates closely with the severity of renal injury, highlighting its role as a potential marker for disease progression.

OPN contributes to kidney injury through multiple mechanisms, primarily by promoting apoptosis in tubular epithelial cells and facilitating the fibrotic response. It activates macrophages and fibroblasts, two key cellular mediators of inflammation and fibrosis, that play central roles in the progression of CKD [105]. In particular, OPN upregulates the production of pro-fibrotic factors such as transforming growth factor-beta (TGF-β), a crucial mediator of fibrosis, and enhances the deposition of extracellular matrix (ECM) components like collagen. This accumulation of ECM leads to tubulointerstitial fibrosis, which disrupts normal kidney architecture, impairing renal function and accelerating disease progression.

Elevated OPN expression in renal tubular cells is a critical driver of fibrosis, as it enhances the synthesis of fibrotic mediators and accelerates ECM deposition. The excessive buildup of ECM contributes to the thickening of kidney tissues, further impairing normal renal function. In addition to promoting fibrosis, OPN also facilitates macrophage infiltration into kidney tissue. Macrophages, which are integral to the inflammatory response in CKD, not only exacerbate inflammation but also support the fibrotic process by releasing cytokines and other mediators that promote ECM production [128]. This interaction between inflammation and fibrosis creates a vicious cycle, driving progressive kidney damage.

Moreover, OPN induces the migration and proliferation of vascular smooth muscle cells (VSMCs), which are involved in the remodeling of blood vessels, particularly in the glomerular and tubular basement membranes. While VSMC migration is necessary for initial tissue repair, excessive VSMC proliferation can lead to pathological vascular remodeling, contributing to a narrowing of renal blood vessels and further ischemic injury [129]. This worsening ischemia impairs kidney perfusion, exacerbating renal dysfunction and contributing to the deterioration of kidney function in CKD.

Studies using animal models of CKD have demonstrated that inhibiting OPN expression or blocking its activity can significantly reduce kidney fibrosis and inflammation. These findings underscore the critical role of OPN in driving renal damage and suggest that targeting OPN could provide a promising therapeutic approach to slow the progression of CKD and mitigate associated fibrosis. Furthermore, OPN levels in serum and urine may serve as valuable biomarkers for assessing the degree of kidney injury and for monitoring the effectiveness of therapeutic interventions aimed at reducing fibrosis and inflammation [106]. By targeting OPN, it may be possible to not only attenuate renal injury but also to prevent or delay the progression to ESRD, offering significant clinical benefits for CKD patients [127].

### 3.7. OPN and Non-Alcoholic Fatty Liver Disease

Non-alcoholic fatty liver disease (NAFLD) is a progressive liver disorder marked by the accumulation of fat in the liver, accompanied by inflammation and fibrosis. NAFLD is strongly linked to metabolic risk factors such as obesity, type 2 diabetes, and dyslipidemia, and if left unmanaged, it can progress to cirrhosis, liver failure, and even hepatocellular carcinoma [107]. The elevated expression of OPN is a consistent finding in the liver tissues of patients with both NAFLD and NASH, reflecting its critical role in disease pathogenesis. OPN is primarily produced by macrophages and hepatic stellate cells (HSCs), which are pivotal players in liver inflammation and fibrosis.

OPN contributes to liver injury by promoting the activation of macrophages and HSCs. In these cells, OPN triggers the release of inflammatory cytokines and fibrogenic mediators, amplifying the inflammatory cascade and stimulating the deposition of extracellular matrix (ECM) components, which drive the fibrotic response. Research has demonstrated that OPN activates key signaling pathways such as TGF-β and NF-κB, both of which are instrumental in the expression of fibrotic factors like collagen and α-smooth muscle actin. These signaling pathways not only enhance ECM deposition but also contribute to the progression of liver fibrosis—a hallmark of NAFLD and NASH. Through its action on ECM remodeling, OPN facilitates the transition from simple steatosis to the more advanced stage of NASH, thus increasing the risk of cirrhosis and associated liver complications [16].

In experimental models of NAFLD, the inhibition of OPN expression has been shown to significantly reduce both liver inflammation and fibrosis, providing strong evidence of the pivotal role of OPN in mediating liver damage. These findings highlight the therapeutic potential of targeting OPN or its signaling pathways as a means to mitigate liver injury, slow fibrosis progression, and improve patient outcomes in NAFLD and NASH. Given its central role in inflammation and fibrosis, OPN stands out as a promising biomarker for disease progression and a potential therapeutic target for modulating the course of NAFLD and NASH. By targeting OPN or its downstream pathways, it may be possible to reduce the risk of liver cirrhosis, halt disease progression, and ultimately improve the long-term prognosis for patients suffering from these increasingly prevalent liver diseases [130].

### 3.8. OPN and Asthma and Chronic Obstructive Pulmonary Disease

Asthma and COPD are chronic respiratory disorders that share a common feature of airway inflammation and airflow limitation, albeit with distinct etiologies and pathophysiological characteristics. Asthma is generally characterized by reversible airflow obstruction and heightened airway responsiveness, often triggered by environmental factors and allergens [108]. In contrast, COPD is a progressive and largely irreversible disease primarily resulting from long-term exposure to harmful particulate matter and gases, such as those found in tobacco smoke. Both diseases substantially impact patients’ quality of life and present significant challenges to public health systems globally [109].

In the context of both asthma and COPD, OPN is increasingly recognized as a key mediator in the underlying pathophysiological processes. Elevated OPN expression has been observed in airway and lung tissues, reinforcing its critical role in the inflammatory and remodeling processes central to the development and progression of these diseases. OPN contributes to airway remodeling and chronic inflammation by promoting the proliferation, migration, and activation of airway smooth muscle cells, which result in airway narrowing, fibrosis, and structural alterations [110]. Furthermore, OPN interacts with immune cells, such as macrophages and neutrophils, facilitating the release of pro-inflammatory cytokines and perpetuating the chronic inflammatory environment characteristic of both asthma and COPD.

Experimental models of asthma have demonstrated that the inhibition or genetic deletion of OPN leads to a reduction in airway inflammation, smooth muscle cell proliferation, and structural remodeling, ultimately improving lung function. These findings underline the central role of OPN in the pathogenesis of these chronic respiratory diseases. Moreover, the involvement of OPN in both inflammation and tissue remodeling in asthma and COPD positions it as a promising therapeutic target. Targeting OPN or its signaling pathways may offer a novel approach to controlling airway inflammation, mitigating remodeling, and ultimately improving clinical outcomes for patients suffering from asthma and COPD [131].

### 3.9. OPN and Tumor Progression and Metastasis

Breast cancer, liver cancer, colorectal cancer, and lung cancer are among the most prevalent and clinically challenging malignancies, characterized by uncontrolled cellular proliferation, evasion of apoptotic mechanisms, and a high propensity for metastatic spread. The progression of these cancers is often driven by perturbations in the tumor microenvironment, which include sustained inflammation, immune evasion, and alterations in the extracellular matrix (ECM) composition [132]. Despite significant advances in early detection and therapeutic interventions, these malignancies remain the leading cause of cancer-related mortality worldwide, with metastasis and therapy resistance being principal contributors to their lethality.

In this regard, OPN has emerged as a critical mediator in the pathophysiology of these cancers. OPN is frequently overexpressed in tumor tissues across a variety of cancer types, and its elevated expression correlates with key aspects of tumor progression, including metastasis, immune suppression, and tissue remodeling. OPN facilitates tumor cell adhesion, migration, and invasion, which are essential processes for metastatic dissemination [133]. By promoting interactions between tumor cells and the ECM, OPN not only supports the detachment of malignant cells from the primary tumor site but also enables their dissemination to distant organs, a hallmark of metastasis [134].

Moreover, OPN exerts profound effects on the immune microenvironment of tumors by interacting with immune cells such as tumor-associated macrophages (TAMs) and myeloid-derived suppressor cells (MDSCs). These immune cells are central to creating an immunosuppressive milieu that allows the tumor to evade immune surveillance. Additionally, OPN promotes neutrophil infiltration into the tumor microenvironment, further sustaining a chronic inflammatory state that supports tumor growth and metastasis. OPN-driven interactions with TAMs and MDSCs inhibit T cell activation and reduce the efficacy of anti-tumor immune responses, further facilitating tumor progression [135].

Beyond its role in immune evasion, OPN is a key driver of chronic inflammation within the tumor microenvironment. OPN also activates the NF-κB signaling pathway, a pivotal mediator of inflammation, which leads to the upregulation of pro-inflammatory cytokines such as interleukin-6 (IL-6) and tumor necrosis factor-alpha (TNF-α) [136]. These cytokines play a crucial role in establishing a pro-inflammatory microenvironment conducive to tumor growth, angiogenesis, and metastatic spread. Additionally, OPN enhances oxidative stress within tumor cells by modulating NADPH oxidase (NOX) activity, leading to increased reactive oxygen species (ROS) production [137]. This oxidative stress contributes to genomic instability, tumor progression, and resistance to apoptosis. In addition, OPN-induced inflammation promotes tissue remodeling and neoangiogenesis, which enhances nutrient supply to the tumor and supports its continuous growth and survival.

Emerging evidence suggests that OPN also contributes to therapeutic resistance in cancer. By activating PI3K/Akt and STAT3 signaling, OPN enhances cancer cell survival and protects against apoptosis induced by chemotherapy or targeted therapies. For example, OPN overexpression has been linked to resistance to platinum-based chemotherapy in lung cancer and colorectal cancer. In contrast, in breast cancer, it has been associated with reduced response to endocrine therapy [138].

Preclinical studies have demonstrated that targeting OPN through monoclonal antibodies or small molecule inhibitors can effectively attenuate tumor growth and metastasis and improve the therapeutic efficacy of other cancer treatments. Notably, OPN-neutralizing antibodies have shown promising anti-tumor effects in preclinical models of breast cancer, colorectal cancer, and melanoma, further validating the therapeutic potential of OPN modulation in cancer treatment [139]. Furthermore, combinatorial strategies targeting OPN along with immune checkpoint inhibitors (e.g., anti-PD-1/PD-L1 therapy) have shown potential in overcoming OPN-driven immune evasion, enhancing anti-tumor immunity.

In summary, osteopontin plays an indispensable role in tumor progression by facilitating tumor cell adhesion, migration, and immune evasion and by modulating the tumor microenvironment. As a key regulator of metastasis, inflammation, and ECM remodeling, OPN represents a promising therapeutic target for inhibiting tumor growth and metastasis, overcoming immune suppression, and enhancing the efficacy of existing cancer therapies. Given its role in tumor-associated inflammation and therapy resistance, future studies should focus on integrating OPN-targeting therapies with existing immunotherapies and chemotherapy regimens to achieve optimal treatment outcomes. Targeting OPN or its downstream signaling pathways may offer novel therapeutic strategies for the management of various cancers, ultimately improving patient outcomes and extending survival in individuals affected by these aggressive malignancies.

## 4. Therapeutic Strategies Targeting OPN in Inflammatory Diseases

As mentioned above, OPN is a key mediator of inflammatory processes, playing a critical role in immune cell recruitment, cytokine production, and tissue remodeling. Its involvement in the pathogenesis of numerous chronic inflammatory diseases has positioned OPN as a promising therapeutic target. As a multifunctional biomolecule, OPN interacts with various cell surface receptors, including integrins and CD44, facilitating the progression of inflammation and tissue damage. Given the central role of OPN in regulating immune responses and tissue homeostasis, several strategies have been developed to modulate its activity, with the goal of alleviating inflammation and improving disease outcomes [140]. These approaches include the use of monoclonal antibodies, small molecule inhibitors, gene silencing techniques, and natural compounds, each aimed at disrupting the interactions of OPN with its receptors or modulating its expression (Figure 4). In the following sections, we will explore these strategies in detail, highlighting their potential as therapeutic interventions in chronic inflammatory diseases.

### 4.1. Monoclonal Antibodies

Neutralizing antibodies against OPN have been explored as a means to block its interaction with integrin receptors, thereby reducing inflammatory cell recruitment and cytokine production. In models of rheumatoid arthritis (RA), it has been demonstrated that OPN deficiency led to decreased joint destruction, highlighting the role of OPN in promoting osteoclastogenesis within inflamed joints [93]. For example, Agnihotri et al. demonstrated that anti-OPN antibodies significantly reduced cartilage erosion and inflammation in a collagen-induced arthritis model, supporting the potential of these antibodies as therapeutic tools for RA [141]. Similarly, in experimental autoimmune encephalomyelitis (EAE), a model of multiple sclerosis (MS), Chabas et al. showed that anti-OPN antibodies mitigated demyelination and inflammation, suggesting a critical role of OPN in neuroinflammatory diseases [142]. These findings collectively suggest that OPN-targeting antibodies could be an effective therapeutic approach in multiple inflammatory diseases. However, while these preclinical results are promising, further clinical trials are necessary to assess the long-term safety and efficacy of OPN-targeting antibodies in human patients.

### 4.2. Small Molecule Inhibitors

Small molecules designed to interfere with binding to receptors, such as CD44 or integrins, have shown promising results in experimental models of arthritis and colitis [133]. For instance, Wang et al. demonstrated that blocking OPN–CD44 interaction in an arthritis model resulted in reduced macrophage infiltration and a significant decrease in pro-inflammatory cytokine levels, which correlated with reduced disease severity [143]. Similarly, Zhu et al. showed that inhibiting the OPN-integrin pathway in a colitis model reduced inflammatory cell infiltration and improved tissue structure in the colon [144]. These findings suggest that small molecule inhibitors can alleviate chronic inflammation through receptor-specific mechanisms, presenting a potential therapeutic avenue for diseases driven by excessive inflammation. Nevertheless, further research is needed to optimize the pharmacodynamics and pharmacokinetics of these inhibitors to make them suitable for clinical use.

### 4.3. Gene Silencing Techniques

RNA interference (RNAi) strategies, including small interfering RNA (siRNA) and antisense oligonucleotides, have been employed to downregulate OPN expression in inflammatory disease models [145]. For example, Diao et al. demonstrated that siRNA targeting OPN significantly decreased kidney inflammation and immune complex deposition in a lupus nephritis model [146]. Additionally, Chen et al. showed that targeting OPN mRNA in a colitis model reduced immune cell infiltration and cytokine production, supporting the efficacy of gene-silencing approaches for managing inflammation in colitis and related conditions [147]. Although these strategies offer promising therapeutic potential, challenges related to delivery methods and minimizing off-target effects must be addressed before RNAi therapies can be widely implemented in clinical settings.

### 4.4. Natural Compounds

Certain natural products have shown potential as modulators of OPN levels or activity, providing alternative therapeutic avenues. Curcumin, a polyphenolic compound found in turmeric, has been found to downregulate OPN expression, reducing inflammatory markers in models of colitis and arthritis [148]. For instance, Sun et al. observed that curcumin administration in a colitis model significantly reduced immune cell recruitment and pro-inflammatory cytokines, demonstrating its potential as a therapeutic agent through OPN inhibition [149]. Resveratrol, another natural compound found in grapes and berries, has been shown to inhibit OPN expression in models of atherosclerosis, where it reduced plaque formation and inflammatory cytokine production, providing evidence for its use in vascular inflammation [150]. These studies highlight the potential of natural compounds in modulating chronic inflammation, suggesting their role in dietary interventions and nutraceutical applications for managing inflammatory diseases.

In conclusion, therapeutic strategies targeting OPN—ranging from monoclonal antibodies to small molecule inhibitors, gene silencing techniques, and natural compounds—hold significant promise for modulating inflammatory pathways implicated in chronic diseases. While preclinical studies have provided compelling evidence of their efficacy, further clinical trials and research are essential to fully establish their therapeutic potential and to optimize their use in human populations.

## 5. Clinical Applications and Challenges

While targeting OPN shows therapeutic promise, its multifaceted role in both physiological and pathological processes presents several challenges that must be considered:

### 5.1. Pleiotropic Functions of OPN

OPN is involved in a range of critical physiological processes, including bone remodeling, wound healing, and immune system regulation. In particular, the role of OPN in matrix formation and mineralization is indispensable for bone health. In bone-related conditions, where OPN contributes to the maintenance of bone structure and integrity, its long-term inhibition could have adverse effects, potentially compromising bone density and strength. In a study by Syn et al., OPN-deficient mice exhibited reduced bone density and impaired fracture healing, highlighting the importance of OPN in bone remodeling. This suggests that therapeutic strategies targeting OPN must be carefully designed to avoid disrupting its beneficial roles in bone health [16]. Therefore, strategies aiming to target OPN should consider the preservation of the positive functions of OPN, particularly in tissues such as bone and in the context of wound healing.

### 5.2. Biomarker Potential of OPN

The utility of OPN as a biomarker for disease monitoring and therapeutic response is gaining recognition. Elevated serum OPN levels have been correlated with disease severity in inflammatory conditions such as rheumatoid arthritis (RA) and systemic lupus erythematosus (SLE), suggesting that OPN may serve as a potential biomarker for disease activity [98]. However, further validation is required to establish the reliability and applicability of OPN as a diagnostic and prognostic marker across a wider spectrum of inflammatory diseases. Roderburg et al. found that elevated serum OPN levels were significantly associated with mortality in critically ill patients, suggesting its potential as a prognostic biomarker. However, the study also emphasized that factors such as age, gender, and comorbidities may influence OPN levels, which may reduce their specificity [114]. Comprehensive studies are needed to confirm its potential for clinical implementation.

### 5.3. Patient Stratification and Personalized Therapies

Not all patients with inflammatory diseases exhibit elevated OPN levels, nor do they respond uniformly to therapies targeting OPN. Evidence suggests that patients with higher baseline OPN levels may benefit more from anti-OPN therapies, underscoring the importance of patient stratification based on individual OPN expression profiles [151]. For example, a preclinical study by Xie et al. demonstrated that in a murine model of rheumatoid arthritis, only mice with high baseline OPN levels showed significant improvement in joint inflammation following anti-OPN therapy, whereas those with low OPN levels exhibited minimal response [152]. This highlights the need for robust biomarkers to identify patients who are most likely to benefit from OPN-targeted interventions. This approach could enhance the efficacy of OPN-targeted treatments, enabling more personalized and effective therapeutic strategies.

### 5.4. Safety and Efficacy Considerations

The safety and long-term efficacy of OPN-targeted therapies warrant thorough investigation. Given the pivotal role of OPN in tissue repair and immune modulation, inhibiting its activity could lead to unintended consequences, such as impaired wound healing or alterations in bone metabolism [153]. Close monitoring of patients undergoing OPN-targeted treatments is essential, with particular attention given to potential side effects related to tissue regeneration. Rigorous dose management and careful consideration of treatment duration are crucial to ensuring the safe use of OPN inhibitors in clinical practice [154]. In a study by Hirano et al., the neutralization of OPN in a sepsis-induced acute lung injury model reduced neutrophil migration and inflammation but also delayed tissue repair, highlighting the dual role of OPN in inflammation [140]. Therefore, future studies should focus on developing OPN inhibitors with a short half-life or conditional activation to minimize off-target effects. Additionally, combination therapies that target both OPN and other inflammatory pathways could mitigate the risks associated with OPN inhibition.

In conclusion, while OPN-targeted therapies hold significant promise for managing inflammatory diseases, careful consideration of their multifaceted role in physiology is essential to avoid unintended consequences. Further research into its therapeutic potential, biomarker utility, and safety profile is crucial for the advancement of OPN-based treatments in clinical settings.

## 6. Conclusions

Osteopontin (OPN) plays a central role in the pathogenesis of various inflammatory diseases and is a key regulator of immune cell activation and migration, cytokine release, as well as influencing apoptosis, tissue remodeling, and fibrosis. This broad involvement makes OPN a key mediator of inflammatory diseases. In these conditions, OPN drives persistent inflammation and leads to tissue damage and disease progression. However, the dual nature of OPN—pro-inflammatory and protective—highlights the complexity of its role in immune response and inflammation. This ambiguity emphasizes the need to carefully specify the target effects of therapeutic interventions aimed at modulating OPN. Therapeutic strategies targeting OPN, including monoclonal antibodies, small molecule inhibitors, gene silencing technologies, and natural compounds, show promise for reducing inflammation and tissue damage by selectively modulating its pathological functions. However, given the important physiological role of OPN, the development of therapeutic approaches that specifically target its pathogenic activity while maintaining its beneficial effects remains a key challenge. Natural compounds that modulate OPN activity offer particular promise as non-invasive therapeutic options, but the detailed pharmacological mechanisms of these agents need to be further explored. Whether targeting OPN with natural plant-derived compounds, synthetic drugs, or biologics, the therapeutic potential of these approaches in the treatment of inflammatory diseases warrants sustained and rigorous investigation. Future studies should focus on elucidating the environment-dependent role of OPN and improving therapeutic strategies to achieve precise and effective modulation of its activity in disease states.

## Figures and Tables

**Figure 1 biology-14-00428-f001:**
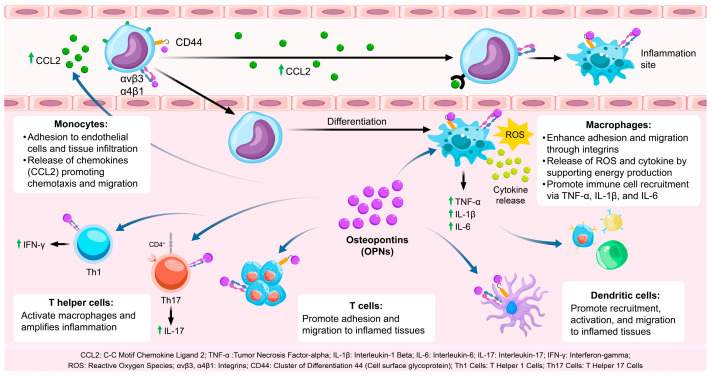
The role of OPN in immune cell recruitment and activation. OPN facilitates immune cell adhesion and migration to inflamed sites via integrins and CD44 and stimulates inflammatory cytokine release through Toll-like receptor activation. CCL2: C-C Motif Chemokine Ligand 2; TNF-α: Tumor Necrosis Factor-alpha; IL-1β: Interleukin-1 Beta; IL-6: Interleukin-6; IL-17: Interleukin-17; IFN-γ: Interferon-gamma; ROS: Reactive oxygen species; αvβ3, α4β1: Integrins; CD44: Cluster of Differentiation 44 (Cell surface glycoprotein); Th1 Cells: T Helper 1 Cells; Th17 Cells: T Helper 17 Cells.

**Figure 2 biology-14-00428-f002:**
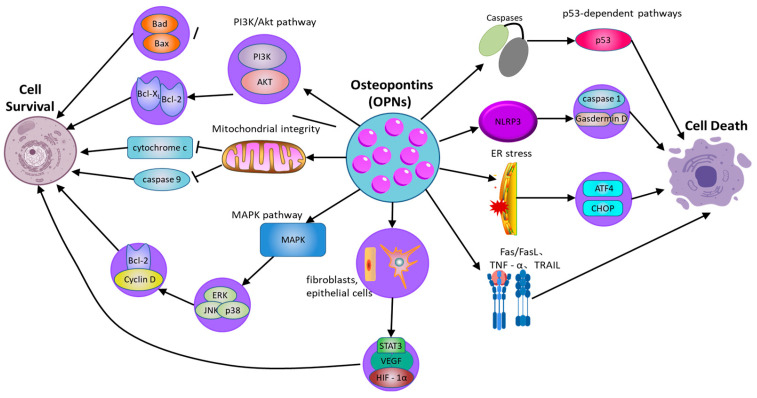
The role of osteopontin in regulating cell survival and cell death pathways. OPN: osteopontin; PI3K/Akt: phosphatidylinositol 3-kinase/protein kinase B; MAPK: mitogen-activated protein kinase; ERK: extracellular signal-regulated kinase; JNK: c-Jun N-terminal kinase; NLRP3: NOD-like receptor family pyrin domain-containing 3; GSDMD: Gasdermin D; ER: Endoplasmic Reticulum; ATF4: Activating transcription factor 4; Fas: Fas cell surface death receptor; FasL: Fas ligand; TNFR: Tumor necrosis factor receptor; TNF-α: Tumor necrosis factor alpha; TRAIL: TNF-related apoptosis-inducing ligand; p53: Tumor protein p53; Caspase-1: Cysteine-aspartic protease 1.

**Figure 3 biology-14-00428-f003:**
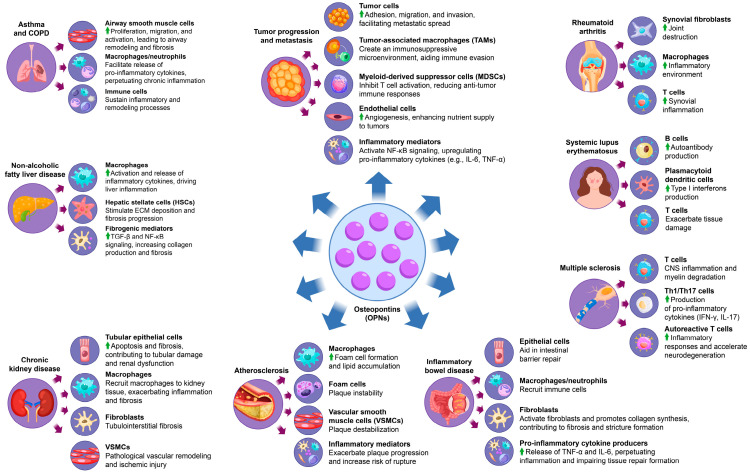
The multifaceted role of OPN in inflammatory disease progression. OPN causes inflammatory disease by affecting a variety of cells and factors, including rheumatoid arthritis, systemic lupus erythematosus, multiple sclerosis, inflammatory bowel disease, atherosclerosis, chronic kidney disease, non-alcoholic fatty liver disease, asthma, and chronic obstructive pulmonary disease and cancer-related inflammation.

**Figure 4 biology-14-00428-f004:**
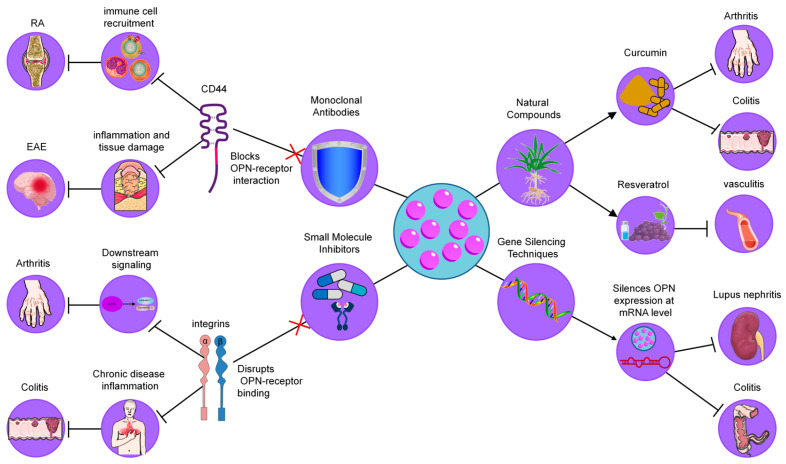
Therapeutic strategies targeting OPN in inflammatory diseases. These strategies include (1) Monoclonal Antibodies, which neutralize OPN by blocking its interaction with integrins and CD44, reducing immune cell recruitment and cytokine production; (2) Small Molecule Inhibitors, designed to disrupt OPN-receptor binding (CD44/integrins), thereby decreasing macrophage infiltration and inflammation; (3) Gene Silencing Techniques, such as siRNA and antisense oligonucleotides, which downregulate OPN expression, leading to reduced immune cell infiltration and cytokine levels; and (4) Natural Compounds, including curcumin and resveratrol, which modulate OPN levels or activity, providing anti-inflammatory effects.

**Table 1 biology-14-00428-t001:** OPN regulates immune cell recruitment and activation.

Immune Cell	Direct Receptor Effects	Downstream Effects	References
Monocytes	Binds to integrins (αvβ3, α4β1) to enhance adhesion to endothelial cells	Recruited to inflammatory sites via CCL2 chemotaxis	[29]
	Interacts with CD44 to promote motility and tissue infiltration	Differentiates into macrophages upon tissue infiltration	[30]
Macrophages	Binds to integrins (αvβ3, α4β1) and CD44 to enhance adhesion and migration	Produces pro-inflammatory cytokines (TNF-α, IL-1β, IL-6) to sustain inflammation	[31]
	Secretes ROS and cytokine	Generates ROS and cytokines to support energy production and inflammatory functions	[32,33]
T cells (Th1)	Secretes IFN-γ to promote adhesion and migration	Activates macrophages and amplifies inflammation	[34]
T cells (Th17)	Secretes IL-17 to promote adhesion and migration	Recruits neutrophils and exacerbates inflammation	[35]
Dendritic cells	Enhances antigen presentation and T cell activation	Promotes recruitment, activation, and migration to inflamed tissues	[36,37]
Immune cell crosstalk	Secretes IL-6, TNF-α, and CCL2	Secrete cytokines and chemokines that activate and recruit T cells, intensifying inflammation	[38]

Abbreviations: OPN: osteopontin; αvβ3: Alpha-v beta-3; α4β1: Alpha-4 beta-1; CD44: Cluster of Differentiation 44; CCL2: C-C Motif Chemokine Ligand 2; TNF-α: Tumor Necrosis Factor Alpha; IL-1β: Interleukin 1 Beta; IL-6: Interleukin 6; ROS: Reactive oxygen species; IFN-γ: Interferon Gamma; IL-17: Interleukin 17; Th1: T helper 1 cells; Th17: T helper 17 cells. References in the table correspond to studies cited in the main text.

**Table 3 biology-14-00428-t003:** Role of osteopontin (OPN) in chronic inflammatory and fibrotic diseases.

Diseases	Function of OPN in the Disease	Promote/Inhibit /Bidirectional	References
Rheumatoid arthritis	OPN promotes inflammation, immune cell activation, and joint destruction	Promote	[90,91,92]
Systemic lupus erythematosus	OPN enhances B cell activation and type I interferon production, driving inflammation	Promote	[95]
Multiple sclerosis	OPN promotes T cell activation and cytokine production, exacerbating CNS inflammation and myelin degradation	Promote	[26,96,97]
Inflammatory bowel disease	OPN supports epithelial repair but also recruits immune cells, exacerbating inflammation and fibrosis	Bidirectional	[82,98]
Atherosclerosis	OPN promotes plaque formation, foam cell accumulation, and VSMC remodeling, contributing to plaque instability and rupture.	Promote	[99,100,101]
Chronic kidney disease	OPN promotes inflammation, fibrosis, and ECM deposition, driving kidney damage and disease progression	Promote	[102,103,104]
Non-alcoholic fatty liver disease	OPN activates macrophages and hepatic stellate cells, promoting inflammation and fibrosis	Promote	[15,105]
Asthma and COPD	OPN promotes airway smooth muscle cell proliferation and inflammation, contributing to airway remodeling	Promote	[106,107]
Breast, liver, colorectal, and lung cancer	OPN promotes tumor cell adhesion, migration, immune evasion, and metastasis, contributing to tumor progression	Promote	[108,109,110]

Abbreviations: OPN: Osteopontin; CNS: Central nervous system; VSMC: Vascular smooth muscle cells; ECM: Extracellular matrix; COPD: Chronic obstructive pulmonary disease. References in the table correspond to studies cited in the main text, providing experimental evidence for the role of OPN in each disease.

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
