# Peer review of "Osteopontin in Chronic Inflammatory Diseases: Mechanisms, Biomarker Potential, and Therapeutic Strategies"

_biology, 2025, doi:10.3390/biology14040428_

Round 1

Reviewer 1 Report

Comments and Suggestions for Authors

In this manuscript, the authors provide a comprehensive review of the multifaceted roles of OPN in immune regulation, inflammation, and tissue remodeling. They delve into the pathogenesis of OPN in various inflammatory diseases and explore potential therapeutic strategies targeting OPN. The review is well-organized, thoroughly examining the biological functions of OPN, its linkage to inflammatory disease progression, and the current developments in therapies targeting OPN. The authors have based their review on a solid compilation of existing studies, presented in an unbiased manner, and have effectively explained the mechanisms of OPN in various biological processes and pathways. Additionally, the manuscript covers a significant number of diseases associated with OPN, reflecting the rigorous work conducted by the authors.

However, there are minor comments that could further improve the manuscript. The challenges of targeting OPN could be more elaborated by providing additional pre-clinical case studies and suggesting possible directions for future research. This would not only enhance the depth of the review but also provide valuable insights for researchers and clinicians working in this field.

Reviewer 2 Report

Comments and Suggestions for Authors

The article by Fuyuan Lang and colleagues provides details of Osteopontin in Chronic Inflammatory Diseases: Mechanisms, Biomarker Potential, and Therapeutic Strategies. In this review article, the authors have mainly highlighted the role of Osteopontin in chronic inflammation, specifically its potential as a biomarker and its therapeutic implications. The article further talks about advancement in preclinical approaches for OPN-driven inflammation. The review delves into the significant role of OPN in modulating inflammation across various immune cells, including macrophages, neutrophils, and T cells. This is a well-written article that provides sufficient details, but the overall quality of the article must be improved, and the following changes should be made.

The figure legends would benefit from additional detail for the Figures. More comprehensive explanations of each figure's content, mechanistic approach, etc., will benefit the reader's interest. 

In section 2.3, OPN regulates cell survival and apoptosis. Here, the authors need to discuss apoptosis in more detail. The authors have discussed the pathways, but the article would benefit from additional discussion along with the addition of a new figure in this section. 

In Table 2. Mechanisms Regulating Apoptosis and Survival Pathways in OPN-Mediated Chronic Inflammation. Here, the authors can further explore apoptosis and its mechanistic action. 

In Figure 2. The multifaceted role of OPN in inflammatory disease progression. The authors should mention the specific types of cells wherever possible. E.g., for macrophages

In section 4. Therapeutic strategies targeting OPN in inflammatory diseases: the authors have discussed multiple approaches focused on disrupting interactions of OPN with its receptors or modulating its expression using monoclonal antibodies, small molecule inhibitors, gene silencing techniques, and natural compounds. The authors provide multiple approaches but fail to discuss them in more detail. The articles will benefit from a more comprehensive literature review in this section. Adding a figure in this section is a must and should reflect the therapeutic strategies targeting OPN. 

The addition of a graphical abstract will further enhance the visibility of the review articles.

Reviewer 3 Report

Comments and Suggestions for Authors

The subject matter of the article is pertinent, however, I have several observations to make.

(1) Section 1. "Introduction". Lines 50-53. In the event of the authors having made reference to inflammasomes, it would be pertinent to indicate not only IL-1 and IL-18, but also pyroptosis.

(2) Section 1. "Introduction". Lines 77-79: it is logical to note that the interaction of OPN with integrins requires limited proteolysis of OPN, including under the influence of thrombin.

(3) The article belongs to the narrative review category. Nevertheless, even in this instance, it is recommended to specify the research methodology, encompassing the databases utilised, the criteria for including and excluding information, and the limitations of the study.

(4) Table 1. The data presented in Table 1 demonstrate a lack of systematic analysis, with data on the direct receptor effects of OPN and the consequences of these effects combined in the same columns. For instance, increased production of cytokines and chemokines in various cells after their activation by OPN is shown. Consequently, this table does not provide a causal relationship analysis. Concurrently, the receptor effects of OPN have the capacity to engender a multitude of equivocal outcomes. It is recommended that the authors utilise this table to systematically categorise the effects of OPN on various cell types. Furthermore, I would advise the authors to conduct a more thorough review of the extant scientific literature, where it is evident that different authors do not always demonstrate the same effects of OPN on the same cell types.

(5) In general, OPN cannot be unambiguously classified as a pro-inflammatory factor, since OPN deficiency can also contribute to low-grade chronic inflammation (see, for example, https://doi.org/10.1038/s41413-024-00355-3). Consequently, it is recommended that the authors approach the analysis of the effect of OPN on various variants of the inflammatory process in a more balanced manner.

(6) The relationship between OPN and the chemokine CCL2 in Figure 1 is not sufficiently clarified.

(7) In Section 2.3, the authors present a unilateral interpretation of the data, failing to acknowledge the ambiguity in classifying OPN as an anti-apoptotic factor (see doi: 10.1152/ajpheart.00954.2013).

(8) Section 3.9: "OPN and tumour progression and metastasis". The objective of the present study, as I understand it, is to demonstrate the correlation between OPN and inflammation. If the authors had chosen to focus on the role of OPN in tumour growth, then it would be logical to also consider the role of pro-inflammatory cellular and tissue stress mechanisms in the development of the tumour process and the influence of OPN on these phenomena.

(9) Section 6 "Conclusion". It is important to note the possibility of ambiguous effects of OPN on the mechanisms of immune response and inflammation, which determines the need to specify the target effects of pathogenetic therapy aimed at OPN.

(10) Literary footnotes are usually given not in round brackets, but in square brackets - [1, 2, 3].

(11) References should be adjusted to the MDPI style and the requirements for the design of articles in the Journal Biology.

Round 2

Reviewer 2 Report

Comments and Suggestions for Authors

The authors have addressed the comments.